# Maternal cytokine profiles in second and early third trimester are not predictive of preterm birth

Kylie K. Hornaday[1], Nikki L. Stephenson[2], Mary T. Canning[3], Suzanne C. Tough[2,3], Donna M. Slater[1,4]*

1 Department of Physiology and Pharmacology, Cumming School of Medicine, University of Calgary, Alberta, Canada, 2 Department of Community Health Sciences, Cumming School of Medicine, University of Calgary, Alberta, Canada, 3 Department of Pediatrics, Cumming School of Medicine, University of Calgary, Alberta, Canada, 4 Department of Obstetrics and Gynecology, Cumming School of Medicine, University of Calgary, Alberta, Canada

☯ These authors contributed equally to this work.
* dmslater@ucalgary.ca

**Data Availability Statement:** The data cannot be made publicly available based on the original ethics approved by the Conjoint Health Region Ethics

## Abstract

Previous studies have investigated whether inflammatory cytokines in maternal circulation are associated with preterm birth. However, many have reported inconsistent results, and few have investigated cytokine trends through gestation, particularly with respect to subtypes of preterm birth. We explored levels of 15 inflammatory cytokines and growth factors in plasma and serum collected in the second (17–23 weeks, timepoint 1 (T1)) and third (28–32 weeks, timepoint 2 (T2)) trimesters with respect to subtypes of preterm birth: spontaneous preterm labour (sPTL), preterm premature rupture of membranes (PPROM), and medically indicated preterm birth (mPTB). The change in TNFα levels over time (T2/T1) significantly classified mPTB from term birth with an AUC of 0.79. While elevated sICAM-1 levels were significantly associated with sPTL, sICAM-1 was not an effective biomarker for prediction. While statistical differences in some biomarkers, such as TNFα and sICAM-1 were found, these are likely not clinically meaningful for prediction. These results did not reveal a relationship between spontaneous labour and circulating maternal inflammatory biomarkers, however, do suggest distinct inflammatory profiles between subtypes of preterm birth.

## Introduction

Preterm birth (PTB), defined as delivery before 37 weeks of gestation, is a leading cause of neonatal morbidity and mortality worldwide, and occurs in approximately 4–16% of all pregnancies [1]. Though its mechanism is not fully understood, spontaneous onset of preterm labour (sPTL) (defined as spontaneous onset of uterine contractions and cervical dilation prior to 37 weeks) has been associated with uterine influx of leukocytes [2, 3], and elevated circulating inflammatory proteins [4], suggesting an association between inflammation and PTB [4, 5]. Indeed, intrauterine infection is a known risk factor for sPTL and preterm premature rupture of membranes (PPROM), both of which often lead to PTB [6–8]. Previous studies investigating the association between elevated levels of pro-inflammatory cytokines and preterm birth, including

Board (CHREB) at the University of Calgary and participant consent, as public availability would compromise patient privacy. However, data is made available on request from the University of Calgary (allourfamilies@ucalgary.ca) following proposal submission and ethics review for researchers who meet the specific criteria for access to confidential data.

**Funding:** The All Our Families study was funded by an Alberta Innovates Interdisciplinary Team Grant #200700595 and Alberta Children's Hospital Foundation. Cytokine analysis was funded by CIHR (Project funding #PJT-173295) the funders had no role in study design, data collection and analysis, decision to publish, or preparation of the manuscript.

**Competing interests:** The authors have declared that no competing interests exist

interleukin-6 (IL-6) [9–19], tumor necrosis factor-alpha (TNFα) [9–11, 13–16, 19], and granulocyte colony-stimulating factor (G-CSF) [12–14, 18, 20, 21] have reported inconsistent results. For example, while elevated G-CSF in maternal blood was found to be associated with increased odds of spontaneous preterm birth in three studies [12, 20, 21], G-CSF was not found to be associated in other studies [14, 18]. Further, most studies have focused on a single time point in pregnancy and have not examined changes in cytokine levels over time [22]. Heterogeneity in preterm birth and its subtypes may impact the associations found between inflammatory markers and PTB, as there is currently no standardized definition for various subtypes of PTB.

Around one third of PTBs are those indicated for maternal or fetal disorders such as pregnancy induced hypertension (PIH), gestational diabetes (GDM) or fetal growth restriction [8]. There is some suggestion that inflammatory biomarkers are associated with such maternal disorders of pregnancy, though the underlying disorders leading to indication for PTB are likely distinct with respect to inflammatory profiles [23]. Recent studies in inflammatory and other molecular biomarkers for PTB have generally classed those cases with PPROM as a subset within spontaneous PTB [24], together making up two thirds of all PTB worldwide [8, 24]. However, whether the two processes (sPTL and PPROM) share aetiology has not been confirmed. The possible mechanisms governing fetal membrane remodelling in membrane rupture likely differ from those governing uterine smooth muscle contractility and cervical dilation. Further, the contribution of inflammation to each condition warrants further investigation. However, especially in the absence of a known infection, the association between inflammation and preterm birth subtypes is not fully understood.

The aim of this study is to profile the inflammatory cytokine and growth factor expression in maternal blood at two timepoints prior to labour onset and investigate any association with preterm birth subtypes. We hypothesize that circulating maternal inflammatory cytokine and growth factor levels will be distinctly associated with sPTL, PPROM and mPTB. By better understanding the role of inflammation and in preterm birth, we can identify potential targets for prevention and treatment of this significant public health issue.

## Methods

### Study participants and outcomes

This was a case-control study nested within the All Our Families (AOF) Pregnancy cohort, a prospective cohort study which collected biological samples from 1867 pregnant women from the Calgary, Canada area [25–27]. Participants were recruited between May 2008 and December 1st 2010 at least 18 years of age and <25 weeks gestation in collaboration with Calgary Laboratory Services. Participants provided informed written consent for blood sample collection, and complete questionnaires including information related to demographics, emotional and physical health. This study was approved by the Conjoint Health Research Ethics Board at the University of Calgary #REB15-0248 Predicting Preterm Birth Study. Questionnaires were completed by participants at 24 weeks gestation (Q1) and 34–36 weeks gestation (Q2). Biological samples were collected at two points in pregnancy, timepoint 1 (T1) at 17–23 weeks gestation and timepoint 2 (T2) 28–32 weeks gestation. Maternal whole blood was centrifuged and collected into heparin tubes for plasma (T1) and serum collection tubes for serum (T2) then stored at -80°C prior to cytokine expression analysis. Samples were kept frozen and thawed only once prior to cytokine analysis. Biological sample tubes were de-identified and re-coded with unique patient IDs. Data is made available on request, from the University of Calgary (allourfamilies@ucalgary.ca) following proposal submission and ethics review for researchers who meet the specific criteria for access to confidential data.

For the present study, participants who delivered live infants were selected and the biological samples were collected from -80˚C storage between April 2$^{nd}$ 2019 and September 19$^{th}$ 2019 for subsequent analysis. Only participants that previously provided biological samples and completed the questionnaires were included in the current study. Of the total n = 108 participants who delivered preterm (<37 weeks gestation), n = 93 had both a serum and plasma sample available, all of which were included for analysis. Of these, n = 46 delivered following spontaneous preterm labour (sPTL), and n = 45 were medically indicated for preterm delivery. The medically indicated group was further divided as those indicated following PPROM (n = 22), and those indicated for other maternal or fetal indications (mPTB) (n = 25). Additionally, n = 413 participants were selected for having uncomplicated full-term deliveries with term spontaneous labour and spontaneous rupture of membranes and had both serum and plasma samples available. Of these, n = 48 were randomly sampled for analysis (term controls). (Fig 1 and S1 File)

## Cytokine analysis

Matched plasma (T1) and serum (T2) samples from n = 141 study participants (n = 46 sPTL, n = 22 PPROM, n = 25 mPTB, and n = 48 term) were analyzed for the following cytokines; eotaxin, granulocyte colony stimulating factor (G-CSF), granulocyte macrophage colony stimulating factor (GM-CSF), interferon gamma (IFNγ), interleukins 1B (IL-1β), 6 (IL-6), 8 (IL-8), and 10 (IL-10), interleukin 1 receptor antagonist (IL-1ra), monocyte chemoattractant protein (MCP-1), macrophage inflammatory protein alpha (MIP-1a), tumor necrosis factor alpha (TNFα), and the growth factor, vascular endothelial growth factor A (VEGF-A) using a custom 13-plex bead-based assay (EveTechnologies, Calgary, Alberta, Canada). In addition, serum amyloid A (SAA) and soluble intercellular adhesion molecule 1 (sICAM-1) were measured using a separate 2-plex assay due to assay probe compatibility (EveTechnologies, Calgary, Alberta, Canada). Serums and plasmas were independently validated to assess potential differences in cytokine level comparing serum and plasma (S1 File).

## Statistical analysis

The relationship between cytokine level and subtypes of preterm birth was determined using a conditional linear regression model to account for subject-specific components, due to the repeated measure design. A linear mixed effects model, also known as a multi-level model or hierarchical linear model, operates similarly to a linear regression, and as such has similar model assumptions. Subject specific components were specified as random effects parameters, to determine which of the cytokines were associated with each of the preterm birth subtypes, as compared to full-term control pregnancies. Birth subtypes and collection time were incorporated into models as fixed effects parameters, along with confounding factors (maternal age, parity, maternal ethnicity, and maternal smoking status) to assess the difference in mean cytokine level between timepoint 1 and timepoint 2, or preterm birth subtypes and term births. Associations with between cytokine levels and preterm birth subtypes were subsequently assessed using receiver operator curve (ROC) analysis. Cytokine levels were log-10 transformed and median-centred prior to ROC analysis. Analysis and visualization were performed in GraphPad Prism 9, Metaboanalyst, and R v4.0.5.

## Results and discussion

### Population characteristics

Four patients (n = 3 sPTL, n = 1 mPTB) delivered prior to the second biological sample collection, and thus only had a T1 plasma sample available for analysis. Four samples (n = 1 T1

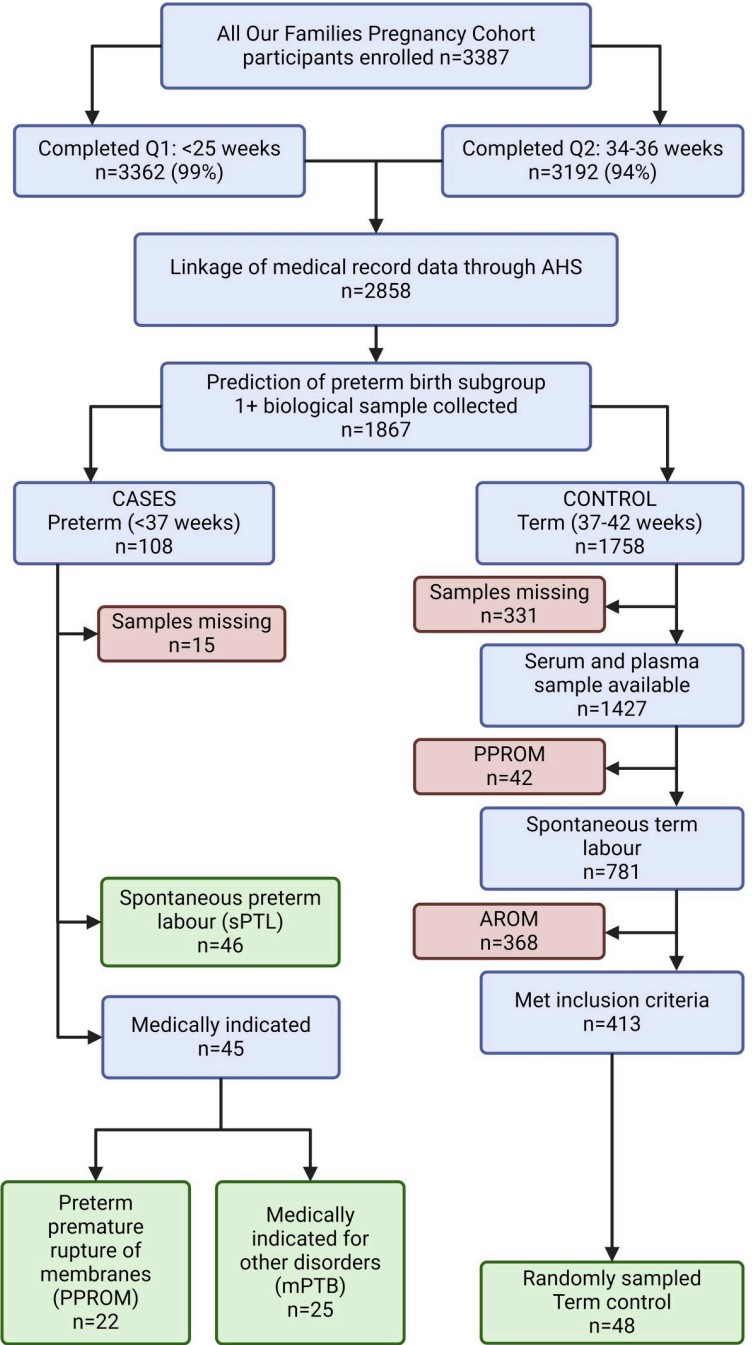

**Fig 1. Diagram of included samples selected from the All Our Families cohort.** Cases were examined in subsets with respect to aetiology of preterm birth. Clinical data linked to the All Our Families database was independently reviewed and discussed by three reviewers (KKH, NLS, DMS) to group cases as 1) preterm births following sPTL, 2) those following preterm premature rupture of membranes (PPROM) with no other symptoms of spontaneous labour 3) preterm deliveries following physician indication (mPTB). An additional 48 subjects who underwent a term (37–40 weeks) birth were included as a control. AROM: artificial rupture of membranes.

plasma, PPROM; n = 1 plasma T1, sPTL; n = 1 T1 plasma, term; and n = 1 T2 serum, term) were not analyzed due to insufficient sample volume for serum/plasma collection. N = 10 T2 serum samples (n = 3 mPTB, n = 2 PPROM, n = 1 sPTL, n = 4 term) were mislabelled or

missing, thus a total of n = 138 plasma and n = 126 serum samples were assessed for cytokine concentration using two custom assays. Demographic characteristics from the study population are presented in Table 1. Participants were predominantly white, attained post-secondary or higher education, married or in common law relationships, and were non-smokers. Consistent with previous studies showing a link with maternal stress and mental health with risk of preterm birth [28, 29], participants with a mPTB reported higher levels of anxiety at Q2, though this was not observed in other subgroups of PTB. It is important to note that Q2 questionnaire data occurred at 34–36 weeks gestation, and thus this association may be confounded with the timing of the preterm deliveries and maternal diagnoses (if relevant). In other words, by 34–36 weeks many with a medically indicated PTB would have received diagnoses for PTB indication (e.g., preeclampsia), which is likely to have an impact on maternal mental health. However, we were unable to collect information on the timing of diagnoses, limiting further investigation of this potential relationship. No differences were found across groups with respect to maternal/paternal age, maternal/paternal ethnicity, education, income, marital status, smoking status, alcohol use during pregnancy, and obstetric history.

## Pregnancy characteristics

Pregnancy characteristics and outcomes are described in Table 2. The sPTL group all delivered following spontaneous onset of labour (63% with PPROM), 4% received additional labour augmentation using oxytocin, and were primarily delivered vaginally (67%). The PPROM group were often induced (91% received oxytocin induction) for a preterm delivery due to advanced gestational age (>34weeks) at time of rupture, or other complications in combination with membrane rupture, and were delivered primarily vaginally (82%). The mPTB group were most likely to be delivered via Caesarean section (64%) with the most common indication for preterm delivery being maternal hypertensive disorders (e.g., 28% preeclampsia, 16% Hemolysis, elevated liver enzymes and low platelets (HELLP syndrome), 16% PIH). All PTB subtypes had significantly shorter gestation and lower birthweight than term infants ($p<0.0001$), as well as longer hospital stays for both the infant and birth parent. The PPROMs had a longer gestation, higher birthweight and shorter hospital stays than the mPTB and sPTL groups. No differences were found in the proportion of male and female infants, APGAR score, instance of small for gestational age (SGA) or large for gestational age (LGA) between preterm and term groups.

## Cytokine and growth factor levels in maternal blood with subtypes of preterm birth

Of the cytokines and growth factors measured, 14 of the 15 were detectable in the study population (Table 3), except for MIP-1a which was below the limit of detection (LOD) in >50% of the population and was thus removed from subsequent analysis. Samples which fell below the LOD were assigned a basement value of 0.64pg/mL. No differences were found in cytokine levels in preterm birth groups as compared to term, at either timepoint.

Separate linear mixed effects modelling was conducted for each cytokine. In a model adjusted for maternal age, maternal ethnicity, parity, smoking status and timepoint of sample collection, levels of sICAM-1 were positively associated with spontaneous preterm labour as compared to term controls (Table 4 and Fig 2). Soluble intercellular adhesion molecule-1 is a constitutively expressed circulating form of ICAM-1 which facilitates leukocyte adhesion and migration across the endothelium [30], and is elevated in all three trimesters of pregnancy as compared to non-pregnant women [31]. While serum and vaginal levels sICAM-1 has been associated with pre-eclampsia (serum only) [31], and PPROM [32], its possible contribution to

**Table 1. Characteristics of study population.**

| | Term (n = 48) | sPTL (n = 46) | PPROM (n = 22) | mPTB (n = 25) |
|---|---|---|---|---|
| **History of miscarriage (%)** | 21[12–34] | 33[21–47] | 18[7–39] | 20[9–39] |
| **History of abortion (%)** | 17[8–30] | 15[8–28] | 0[0–15] | 12[4–30] |
| **History of stillbirth (%)** | 0[0–7] | 4[1–15] | 5[0.8–22] | 0[0–13] |
| **Previous PTB (%)** | 6[2–17] | 20[11–33] | 14[5–33] | 8[2–25] |
| **Family history of PTB (%)** | 15[7–27] | 17[9–31] | 14[5–33] | 4[0.7–20] |
| **Marital status[a] (%)** | 96[86–99] | 93[82–98] | 91[72–97] | 92[75–98] |
| **Education (%)** | | | | |
| high school or less | 4[1–14] | 13[6–26] | 18[7–39] | 12[4–30] |
| some or complete post-secondary | 79[66–88] | 72[59–85] | 59[39–77] | 76[55–91] |
| some or complete graduate | 17[9–30] | 13[6–26] | 23[10–43] | 12[4–30] |
| data not available | -- | 2[0.05–12] | -- | -- |
| **Household Income[b] (%)** | 13[6–25] | 7[2–18] | 23[10–43] | 16[6–35] |
| **Maternal Ethnicity (%)** | | | | |
| White | 79[66–88] | 72[59–85] | 68[47–84] | 68[48–83] |
| Asian | 17[9–30] | 9[3–20] | 9[3–28] | 16[6–35] |
| Other | 4[0.5–14] | 15[6–29] | 23[8–45] | 16[6–35] |
| data not available | -- | 4[0.5–15] | -- | -- |
| **Paternal Ethnicity (%)** | | | | |
| White | 75[61–85] | 76[62–86] | 73[52–87] | 84[65–94] |
| Asian | 17[9–30] | 4[1–15] | 9[3–28] | 8[2–25] |
| Other | 8[2–17] | 17[8–31] | 18[5–40] | 8[2–25] |
| data not available | -- | 2[0.05–12] | -- | -- |
| **Alcohol use * (%)** | | | | |
| non-drinker | 19[9–33] | 13[5–26] | 5[0.8–22] | 4[0.7–20] |
| None | 25[13–37] | 33[20–48] | 50[31–69] | 44[27–63] |
| Low | 48[34–62] | 39[25–55] | 18[5–40] | 32[17–52] |
| moderate | 2[0.3–11] | 2[0.05–12] | 5[0.8–22] | 0[0–13] |
| High | 0[0–7] | 0[0–8] | 5[0.8–22] | 8[2–25] |
| data not available | 6[1–17] | 13[5–26] | 18[5–40] | 12[3–31] |
| **Smoker* (%)** | 15[7–27] | 13[6–26] | 23[10–43] | 16[6–35] |
| **Maternal Age (years)** | 32[31–33] | 32[30–33] | 31[29–34] | 32[30–34] |
| **Paternal Age (years)** | 34[33–35] | 34[33–36] | 34[32–37] | 34[31–37] |
| **Gravidity (no.)** | 2.0[1.7–2.7] | 2.2[1.8–2.6] | 1.7[1.3–2.0] | 1.8[1.4–2.2] |
| **Parity (no.)** | 0.5[0.3–0.7] | 0.7[0.4–0.9] | 0.5[0.2–0.9] | 0.4[0.2–0.6] |
| **Anxiety (score)** | | | | |
| Q1 (24 weeks) | 31[29–33] | 32[29–34] | 31[28–34] | 37[32–42] |
| Q2 (34–36 weeks) | 31[29–34] | 37[33–40] | 36[32–40] | **41[35–48]** |
| **Depression (score)** | | | | |
| Q1 | 4.7[3.8–5.7] | 5.6[4.5–6.7] | 6.2[4.2–8.1] | 7.4[4.9–9.9] |
| Q2 | 4.4[3.1–10] | 5.7[4.2–7.1] | 6.0[4.0–7.9] | 8.2[5.3–11.0] |

Values represented as %[95% confidence interval] for categorical variables or mean[95% confidence interval] for numerical variables. Bold values indicate significantly different than term group.

*after pregnancy recognition.

[a] married or common law.

[b] $60k/year or less.

**Table 2. Pregnancy outcomes and characteristics of study population.**

| | Term (n = 48) | sPTL (n = 46) | PPROM (n = 22) | mPTB (n = 25) |
|---|---|---|---|---|
| **Mode of delivery (%)** | | | | |
| vaginal | 98[89–100] | **67[53–79]** | 82[61–93] | **36[20–55]** |
| emergency CS | 2[0.4–11] | **30[19–45]** | 14[5–33] | **44[27–63]** |
| elective CS | 0[0–7] | 2[0.4–11] | 5[0.8–22] | **20[9–39]** |
| **Labour (%)** | | | | |
| spontaneous | 88[70–91] | **37[25–51]** | NA | NA |
| spontaneous with ROM | 2[0.4–11] | **63[49–75]** | NA | NA |
| ROM only | 10[5–22] | NA | 100 | NA |
| induced | NA | NA | NA | 52[33–70] |
| none | NA | NA | NA | 48[30–67] |
| **Indication for PTB (%)** | | | | |
| sPTL | NA | 100 | NA | NA |
| PPROM | | | | |
| >34weeks | NA | NA | 50[31–69] | NA |
| twins | NA | NA | 5[0.8–22] | NA |
| complications | NA | NA | 45[27–65] | NA |
| preeclampsia | NA | NA | NA | 28[14–48] |
| PIH | NA | NA | NA | 16[6–35] |
| IUGR | NA | NA | NA | 16[6–35] |
| HELLP | NA | NA | NA | 16[6–35] |
| oligohydramnios | NA | NA | NA | 8[2–25] |
| other | NA | NA | NA | 16[6–35] |
| **Induction/Augmentation (%)** | | | | |
| oxytocin | 13[6–25] | 4[1–14] | **91[72–97]** | **52[33–70]** |
| AROM | 2[0.4–11] | 0[0–7] | 14[5–33] | **32[17–52]** |
| none | 85[73–93] | 96[85–99] | **14[5–33]** | **48[30–67]** |
| **Reason for CS (%)** | | | | |
| NA | 98[89–100] | 67[53–79] | 73[52–87] | 32[17–52] |
| FHR anomaly | 0[0–7] | 7[2–18] | 5[0.8–22] | 16[6–35] |
| previous CS | 0[0–7] | 11[5–23] | 5[0.8–22] | 8[2–25] |
| breech | 0[0–7] | 15[8–28] | 9[3–28] | 12[3–31] |
| preeclampsia | 0[0–7] | 0[0–7] | 0[0–15] | 8[2–25] |
| other/unknown | 2[0.4–11] | 0[0–7] | 9[3–28] | **24[9–45]** |
| **Pregnancy complications (%)** | | | | |
| PIH | 4[1–14] | 9[3–20] | **77[57–90]** | 68[48–83] |
| FHR anomaly | 17[9–30] | 11[5–23] | 23[10–43] | 20[9–39] |
| bleeding | 8[3–20] | 9[3–20] | 14[5–33] | 12[4–30] |
| GDM | 2[0.4–11] | 7[2–18] | 0[0–15] | 0[0–13] |
| GBS | 31[20–45] | 9[3–20] | 5[0.8–22] | 4[0.7–20] |
| proteinuria | 0[0–7] | 7[2–18] | **45[27–65]** | **40[23–59]** |
| oligohydramnios | 0[0–7] | 2[0.4–11] | 18[7–39] | 16[6–35] |
| HELLP | 0[0–7] | 0[0–7] | 18[7–39] | 16[6–35] |
| IUGR | 0[0–7] | 0[0–7] | 18[7–39] | 16[6–35] |
| preexisting chronic condition | 2[0.4–11] | 2[0.4–11] | 5[0.8–22] | 4[0.7–20] |
| **Fetal sex (% female)** | 42[29–56] | 30[19–45] | 41[23–61] | 44[27–63] |
| **Fetal presentation (%)** | | | | |

*(Continued)*

**Table 2.** (Continued)

|  | Term (n = 48) | sPTL (n = 46) | PPROM (n = 22) | mPTB (n = 25) |
|---|---|---|---|---|
| vertex | 94[83–98] | **70[55–81]** | 86[67–95] | **52[33–70]** |
| breech | 0[0–7] | **20[11–33]** | 9[3–28] | **24[11–43]** |
| unknown | 6[2–17] | 9[2–21] | 5[0.8–22] | 24[11–43] |
| **Multiple gestation (%)** | 0[0–7] | 7[2–18] | 5[0.8–22] | 16[6–35] |
| **Baby time in hospital (days)** | 1.5[1.1–1.9] | **22.0[14.4–29.7]** | **5.8[4.0–7.5]** | **20.4[12.0–32.1]** |
| **Parent time in hospital (days)** | 1.4[1.2–1.6] | **2.4[1.9–2.8]** | **2.6[1.8–3.3]** | **3.8[2.3–5.3]** |
| **Birthweight (g)** | 3,372[3,242–3,501] | **2,294[2,122–2,466]** | **2,729[2,575–2,893]** | **2,085[1,834–2,336]** |
| **Cervical dilation (cm)** | 4.6[3.8–5.4] | 3.5[2.8–4.2] | **0.8[0.3–1.3]** | **0.6[0.2–1.0]** |
| **5min APGAR (score)** | 8.9[8.8–9.0] | 8.4[8.1–8.8] | 8.8[8.3–9.2] | 8.6[8.3–8.8] |
| **SGA (%)** | 13[6–25] | 2[0.4–11] | 0[0–15] | 24[11–43] |
| **LGA (%)** | 2[0.4–11] | 11[5–23] | 5[0.8–22] | 0[0–13] |
| **GA at delivery (weeks)** | 39.1[38.8–39.4] | **33.2[32.3–34.1]** | **35.3[34.9–35.6]** | **34.3[33.4–35.2]** |
| **Antenatal steroid use (%)** | 0[0–7] | **28[17–43]** | 9[3–28] | **36[20–55]** |

Values represented as %[95% confidence interval] for categorical variables or mean[95% confidence interval] for numerical variables. HIV; human immunodeficiency virus, GBS; group B streptococcus, PIH; pregnancy induced hypertension, GDM; gestational diabetes mellitus, IUGR; intrauterine growth restriction, AROM; artificial rupture of membranes, CS; Caesarean section, SGA; small for gestational age, LGA; large for gestational age. Bold values indicate significantly different from term group.

uterine contractions in spontaneous labour remains to be explored. However, sICAM-1-mediated leukocyte invasion and migration may play a role in the inflammatory phenotype and uterine leukocyte influx seen with labour onset [2, 3, 33].

**Table 3. Cytokine and growth factor levels in maternal blood across pregnancy outcomes.**

|  |  | Term | | sPTB | | PPROM | | mPTB | |
|---|---|---|---|---|---|---|---|---|---|
|  |  | T1 (n = 47) | T2 (n = 43) | T1 (n = 45) | T2 (n = 42) | T1 (n = 21) | T2 (n = 20) | T1 (n = 25) | T2 (n = 21) |
| **SAA** | ug/mL | 8.5 [6.5–10.4] | 6.4 [5.2–7.6] | 7.8 [6.3–9.3] | 6.8 [5.4–8.1] | 8.3 [5.6–11.0] | 7.3 [5.8–8.8] | 6.5 [5.2–7.7] | 5.7 [4.3–7.1] |
| **sICAM1** |  | 0.2 [0.2–0.3] | 0.3 [0.2–0.3] | 0.3 [0.2–0.4] | 0.4 [0.3–0.4] | 0.2 [0.2–0.3] | 0.2 [0.2–0.3] | 0.2 [0.2–0.3] | 0.3 [0.2–0.4] |
| **Eotaxin** |  | 151.4 [131.5–171.3] | 34.3 [29.5–39.1] | 135.1 [116.8–153.3] | 34.9 [30.2–39.6] | 131.9 [108.8–154.9] | 31.1 [20.6–41.5] | 136.8 [108.7–164.8] | 35.6 [25.8–45.4] |
| **G-CSF** |  | 46.2 [39.8–52.6] | 38.2 [33.4–43.0] | 47.1 [36.8–57.5] | 45.1 [35.8–54.4] | 38.7 [30.3–47.1] | 34.0 [25.5–42.4] | 55.3 [30.1–80.6] | 71.7 [32.4–110.9] |
| **GM-CSF** |  | 10.7 [5.8–15.7] | 9.1 [4.5–13.7] | 10.6 [6.8–14.3] | 7.4 [4.7–10.1] | 5.3 [-4.9–15.4] | 3.6 [-4.0–11.3] | 13.8 [1.9–25.6] | 17.0 [-1.3–35.3] |
| **IFNγ** |  | 10.3 [4.2–16.4] | 8.5 [3.1–13.8] | 10.6 [4.9–16.3] | 6.6 [4.3–9.0] | 1.9 [-0.7–4.5] | 1.9 [-4.7–8.4] | 7.5 [0.9–14.2] | 11.6 [2.5–20.7] |
| **IL-1β** | pg/mL | 2.9 [1.8–3.9] | 2.6 [1.5–3.7] | 3.8 [2.1–5.5] | 2.6 [1.4–3.7] | 1.2 [-1.8–4.2] | 0.8 [-3.8–5.5] | 3.7 [-0.2–7.5] | 3.6 [-0.9–8.1] |
| **IL-1ra** |  | 45.7 [36.6–54.7] | 40.7 [30.7–50.7] | 37.9 [30.7–45.0] | 43.2 [32.9–53.6] | 26.1 [12.4–39.7] | 27.9 [12.5–43.4] | 45.1 [29.2–61.0] | 57.4 [38.5–76.3] |
| **IL-6** |  | 5.0 [2.5–7.6] | 5.0 [2.4–7.7] | 5.7 [3.0–8.4] | 5.8 [3.2–8.3] | 1.3 [-1.6–4.3] | 1.5 [-1.1–4.1] | 5.0 [0.2–9.8] | 3.8 [-0.5–8.1] |
| **IL-8** |  | 12.4 [6.9–17.9] | 32.8 [2.1–63.4] | 11.5 [6.9–16.1] | 21.1 [4.4–37.8] | 6.5 [1.2–11.9] | 8.7 [-9.8–27.2] | 19.9 [-1.8–41.7] | 21.3 [-1.7–44.3] |
| **IL-10** |  | 3.0 [2.2–3.9] | 2.5 [1.8–3.1] | 5.9 [0.7–11.1] | 6.0 [1.3–10.6] | 1.6 [-0.3–3.5] | 1.5 [-2.9–6.0] | 3.4 [1.1–5.8] | 4.0 [0.9–7.2] |
| **MCP-1** |  | 221.4 [194.2–248.7] | 251.2 [219.0–283.5] | 236.4 [200.4–272.5] | 286.2 [240.9–331.6] | 219.3 [189.8–248.7] | 221.5 [168.1–274.9] | 226.6 [195.3–257.9] | 267.6 [223.1–312.1] |
| **TNFα** |  | 11.0 [9.1–12.9] | 9.9 [8.5–11.4] | 11.3 [9.1–13.5] | 11.5 [9.5–13.5] | 8.5 [6.1–10.9] | 8.9 [4.8–12.9] | 11.2 [6.6–15.8] | 15.6 [10.2–21.0] |
| **VEGF-A** |  | 10.7 [7.8–13.7] | 11.0 [7.1–14.8] | 15.6 [9.9–21.2] | 13.3 [9.2–17.3] | 4.8 [1.9–7.7] | 5.1 [0.4–9.8] | 23.8 [-4.2–51.8] | 26.5 [-1.5–54.5] |

Levels are reported as mean with 95% confidence interval.

**Table 4. Linear mixed effects analysis of cytokine levels across preterm birth subtypes.**

| | | Crude RR | | | | Adjusted RR | | |
|---|---|---|---|---|---|---|---|---|
| | Time | sPTL | PPROM | mPTB | Time | sPTL | PPROM | mPTB |
| SAA | **-1.30E+06** | -2.70E+05 | 1.70E+06 | -1.40E+06 | **-1.30E+06** | -3.90E+05 | 2.60E+06 | -1.20E+06 |
| | **[-2.0e+06,-5.5e+05]** | [-2.1e+06, 1.6e+06] | [-6.3e+05, 4.0e+06] | [-3.6e+06, 8.9e+05] | **[-2.1e+06,-5.6e+05]** | [-2.3e+06, 1.5e+06] | [-1.1e+04, 5.2e+06] | [-3.5e+06, 1.1e+06] |
| sICAM1 | 4.00E+04 | 6.00E+04 | 6474.353 | 1012.744 | 4.00E+04 | 7.50E+04 | -1.10E+04 | -1.20E+04 |
| | **[1.5e+04, 6.6e+04]** | [-3.6e+03, 1.2e+05] | [-7.3e+04, 8.6e+04] | [-7.6e+04, 7.8e+04] | **[1.3e+04, 6.6e+04]** | **[1.1e+04, 1.4e+05]** | [-9.7e+04, 7.6e+04] | [-8.8e+04, 6.4e+04] |
| Eotaxin | **-107.489** | -9.133 | -2.553 | -7.018 | **-107.505** | -9.124 | 0.878 | -8.713 |
| | **[-117.884,-97.093]** | [-25.265,6.999] | [-22.747,17.641] | [-26.424,12.388] | **[-118.431,-96.579]** | [-25.143,6.895] | [-20.973,22.729] | [-27.997,10.570] |
| G-CSF | -2.293 | 3.138 | -0.637 | 18.37 | -1.984 | 2.779 | -2.635 | 16.098 |
| | [-6.170,1.583] | [-12.388,18.664] | [-20.022,18.749] | [-0.212,36.953] | [-6.170,2.202] | [-13.422,18.980] | [-24.512,19.242] | [-3.210,35.407] |
| GM-CSF | **-1.788** | -0.702 | 1.179 | 4.961 | **-1.721** | -0.964 | -2.863 | 4.982 |
| | **[-3.375,-0.200]** | [-8.768,7.363] | [-8.889,11.248] | [-4.688,14.610] | **[-3.306,-0.136]** | [-9.008,7.081] | [-13.715,7.989] | [-4.596,14.560] |
| IFNγ | -0.038 | 0.162 | -3.077 | -0.483 | -0.095 | -0.675 | -1.372 | 0.074 |
| | [-1.940,1.865] | [-6.672,6.995] | [-11.611,5.456] | [-8.664,7.698] | [-2.158,1.967] | [-7.910,6.561] | [-11.148,8.403] | [-8.553,8.701] |
| IL-1B | -0.469 | 0.486 | 0.848 | 0.875 | -0.475 | 0.402 | 1.078 | 0.744 |
| | [-1.039,0.100] | [-2.042,3.013] | [-2.307,4.003] | [-2.150,3.899] | [-1.092,0.141] | [-2.286,3.091] | [-2.550,4.707] | [-2.459,3.947] |
| IL-1ra | 2.489 | -3.592 | -0.629 | 7.36 | 3.156 | -5.128 | -8.571 | 4.037 |
| | [-2.219,7.197] | [-16.238,9.054] | [-16.429,15.172] | [-7.796,22.516] | [-1.740,8.051] | [-17.923,7.667] | [-25.893,8.752] | [-11.250,19.323] |
| IL-6 | 0.005 | 0.733 | -0.288 | 0.241 | -0.115 | 0.39 | -0.063 | 0.098 |
| | [-0.422,0.433] | [-2.980,4.446] | [-4.922,4.346] | [-4.199,4.680] | [-0.530,0.299] | [-3.564,4.344] | [-5.391,5.266] | [-4.605,4.802] |
| IL-8 | 11.918 | -5.965 | -6.197 | -1.182 | 11.064 | -6.817 | -11.37 | -1.984 |
| | [-0.143,23.978] | [-23.295,11.365] | [-27.898,15.503] | [-22.042,19.679] | [-1.660,23.788] | [-24.955,11.321] | [-36.122,13.383] | [-23.828,19.861] |
| IL-10 | 0.209 | 3.009 | 1.73 | 0.871 | -0.095 | 3.294 | 0.612 | 0.618 |
| | [-0.511,0.929] | [-1.076,7.094] | [-3.369,6.829] | [-4.015,5.757] | [-0.723,0.532] | [-0.950,7.539] | [-5.110,6.334] | [-4.433,5.669] |
| MCP-1 | **36.594** | 13.47 | 5.01 | 4.856 | **34.411** | 15.958 | 3.732 | -6.405 |
| | **[23.094,50.093]** | [-26.854,53.794] | [-45.474,55.493] | [-43.213,52.925] | **[21.938,46.885]** | [-26.268,58.184] | [-53.523,60.987] | [-56.606,43.795] |
| TNFα | 0.569 | 0.76 | -0.068 | 2.333 | 0.356 | -0.13 | -1.14 | 2.075 |
| | [-0.466,1.604] | [-2.286,3.805] | [-3.873,3.736] | [-1.316,5.982] | [-0.724,1.435] | [-3.184,2.925] | [-5.273,2.993] | [-1.572,5.722] |
| VEGF-A | 0.14 | 4.182 | -1.835 | 13.085 | 0.149 | 3.541 | -1.366 | 13.067 |
| | [-1.923,2.203] | [-8.011,16.376] | [-17.055,13.385] | [-1.500,27.670] | [-2.089,2.386] | [-9.459,16.541] | [-18.896,16.165] | [-2.407,28.541] |

The difference in mean cytokine level between preterm birth subtypes and term births, and between T1 and T2 (adjusted for subject specific intercepts and other maternal factors) are reported with [95% confidence intervals]. Cytokine levels for SAA and sICAM-1 were transformed by centering around the minimum value prior to analysis. Bold values indicate significantly non-zero.

## Changes in cytokine and growth factor levels with gestational age

Levels of SAA, eotaxin, GM-CSF, sICAM-1 and MCP-1 were observed to change across time-points in all groups (Fig 3). No significant differences between timepoints were observed in any other biomarker measured. SAA, eotaxin and GM-CSF were observed to be lower at T2 as compared to T1, which may suggest perturbations in leukocyte activation/production through pregnancy. GM-CSF plays an important role in hematopoiesis by promoting differentiation of stem cells to granulocytes and antigen presenting cells, specifically basophils, neutrophils, eosinophils, and monocytes [34]. Though the function of SAA is not well understood, it likely plays a role in acute phase response [35]. Eotaxin demonstrated the most substantial decrease with gestational age. This dramatic decrease in eotaxin levels was also reported in a Norwegian

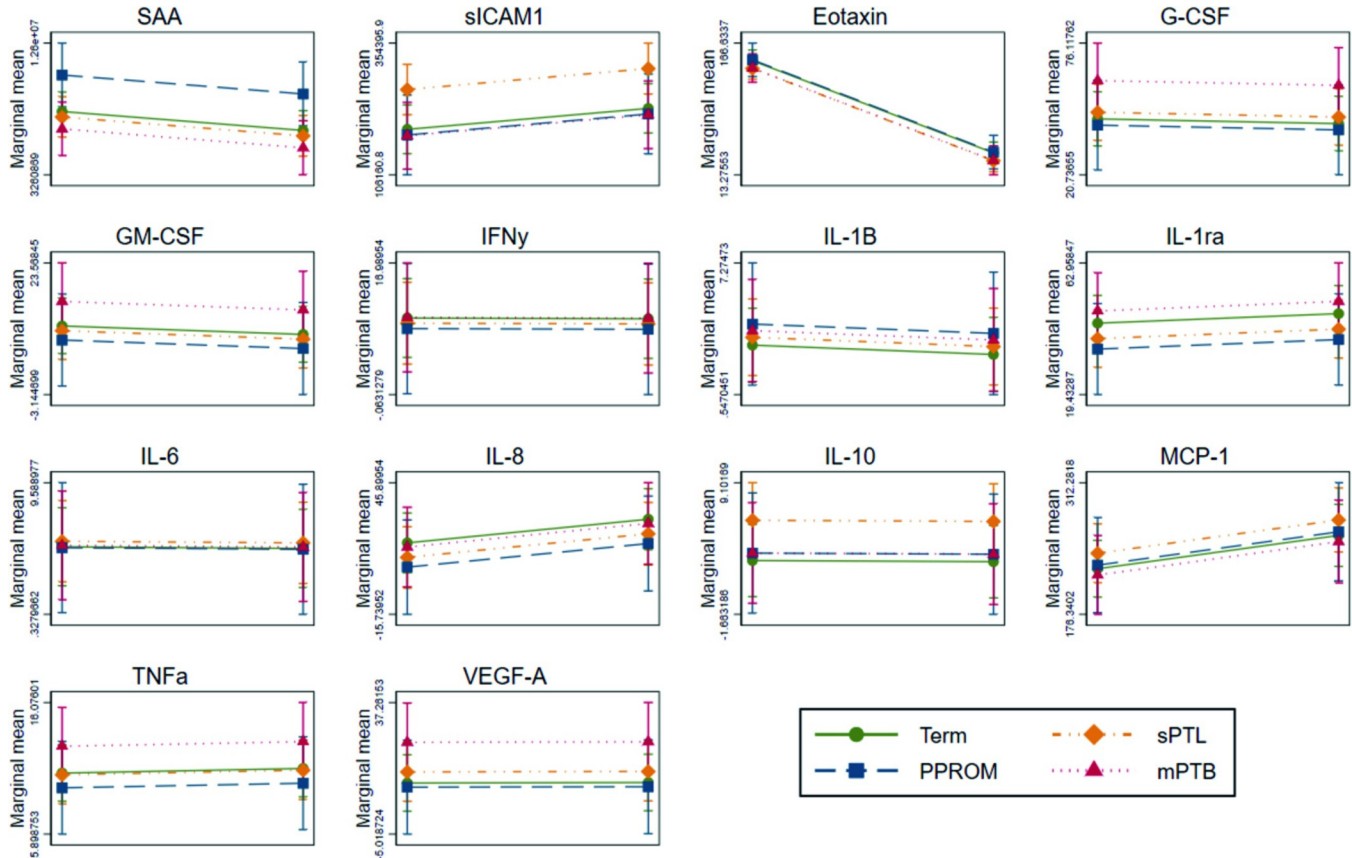

**Fig 2. Linear mixed effects analysis of two timepoints in different subtypes of preterm birth.** Figures were produced using the marginal model error variance structure at two timepoints in pregnancy, showing marginal mean levels of cytokine (y-axis), across two timepoints in pregnancy (x-axis).

pregnancy cohort [36], though the precise regulatory mechanisms and function remain unclear. Eotaxin is involved in eosinophil recruitment, which has been shown to be important for endometrial remodelling in early pregnancy in mice [37], and that eosinophil recruitment in the mouse uterus is eotaxin-dependent [38]. However, the role of eotaxin and eosinophils in human pregnancy and labour is not well understood, though high levels of eosinophils in amniotic fluid have been associated with sPTL [39], and low eosinophil count has been

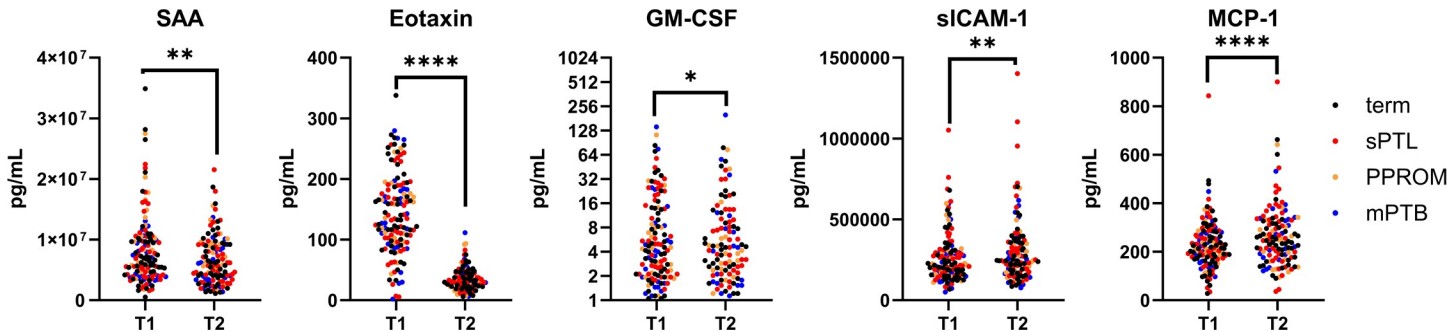

**Fig 3. Cytokine levels across timepoints.** Time was assessed as fixed effect parameter to determine whether it significantly contributed to the relationship between cytokine level and preterm birth using a linear mixed effects model. *$p < 0.05$, **$p < 0.01$, ****$p < 0.0001$. SAA; serum amyloid A, sICAM-1; soluble intercellular adhesion molecule 1, GM-CSF; granulocyte macrophage colony stimulating factor, MCP-1; monocyte chemoattractant protein.

implicated in preeclampsia [40]. It is hypothesized that the decidual stromal cells play a predominant role in secreting cytokines during term labour to activate uterine microvascular endothelial cells to increase migration of leukocytes to the uterine environment [41], however, whether these cytokines are detectable in maternal peripheral blood or other biological tissue remains to be seen. Further investigation of cytokine and immune cell levels through more timepoints, specifically earlier than 17 weeks, after 32 weeks, and with comparison to non-pregnant levels may provide more insight to the physiological changes that occur during pregnancy but may not yield information about the risk of preterm birth. On the other hand, sICAM-1 and MCP-1 (monocyte chemoattractant protein -1) levels were higher at T2 compared to T1, which may suggest increases in the recruitment of leukocytes, such as monocytes, and other inflammatory molecules during pregnancy.

## Group specific interactions

Models were additionally assessed to determine a possible interaction between birth group (sPTL, PPROM, mPTB or term) and timepoint of collection. We found no significant interactions between the groups and time for SAA, sICAM-1, eotaxin, GM-CSF, IFNγ, IL-1B, IL6, IL8, MCP-1 and VEGF-A, suggesting that there were no significant differences in slope between timepoints among groups, therefore the interaction was removed. Both IL-1ra and TNFα exhibited no significant change between timepoints in all groups (slope not significantly different from zero) except mPTB, which exhibited a positive slope (slope[95% confidence interval] IL-1ra: 19.239[4.471, 32.007], TNFα: 4.573[1.536, 7.61]). Mean levels of G-CSF within sPTL and PPROM groups did not change significantly over time, yet levels decreased significantly within the term group (-8.329[-15.029, -1.629]) and increased significantly in the mPTB group (20.923[9.325, 32.52]) This suggests that levels of IL-1ra, TNFα and G-CSF may increase over time in those who subsequently deliver following medical indication for PTB, but not in other etiologies of PTB. However, further investigation of the role of these cytokines in the distinct etiologies of mPTB, commonly preeclampsia, other hypertensive disorders of pregnancies and/or fetal growth restrictions would be required to further elucidate this relationship and possible role as biomarkers. On the other hand, mean levels of IL-10 remain significantly unchanged between timepoints for all groups except PPROM, which exhibited a positive slope (3.369[1.24, 5.498]). Summarized in Fig 4. Full table of slopes for interaction terms can be found in S1 Table.

## Predictive power of cytokines and other risk factors for preterm birth

Biomarker analysis was conducted using univariate receiver operating characteristic (ROC) curve analysis. For mPTB, TNFα T2/T1 ratio exhibited the strongest classification with an AUC of 0.79, followed by G-CSF T2/T1 ratio with an AUC of 0.77, Q1 anxiety score with an AUC of 0.65, and G-CSF levels at T2 with AUC of 0.65. Combined biomarker analysis of TNFα ratio/Q1 anxiety score performed similarly with an AUC of 0.67 (Fig 5). Other biomarkers did not exhibit classification with an AUC significantly greater than 0.5 (random classification), for any subtype of PTB. TNFα is an inflammatory cytokine produced mainly by T-lymphocytes as part of the inflammatory response. It acts via two TNF receptors, TNFR1 and TNFR2, to provoke a range of cellular responses, including apoptosis, inflammation, proliferation and tissue repair [42]. Of the subjects included in this study, most mPTBs were indicated for maternal hypertensive disorders (60%). Hypertension is characterized by low grade inflammation, including the presence of TNFα, and has been implicated in kidney function and kidney injury repair [43]. Endogenous treatment with TNFα induces significant increase in blood pressure in pregnant rats as compared to non-pregnant rats and is associated with activation

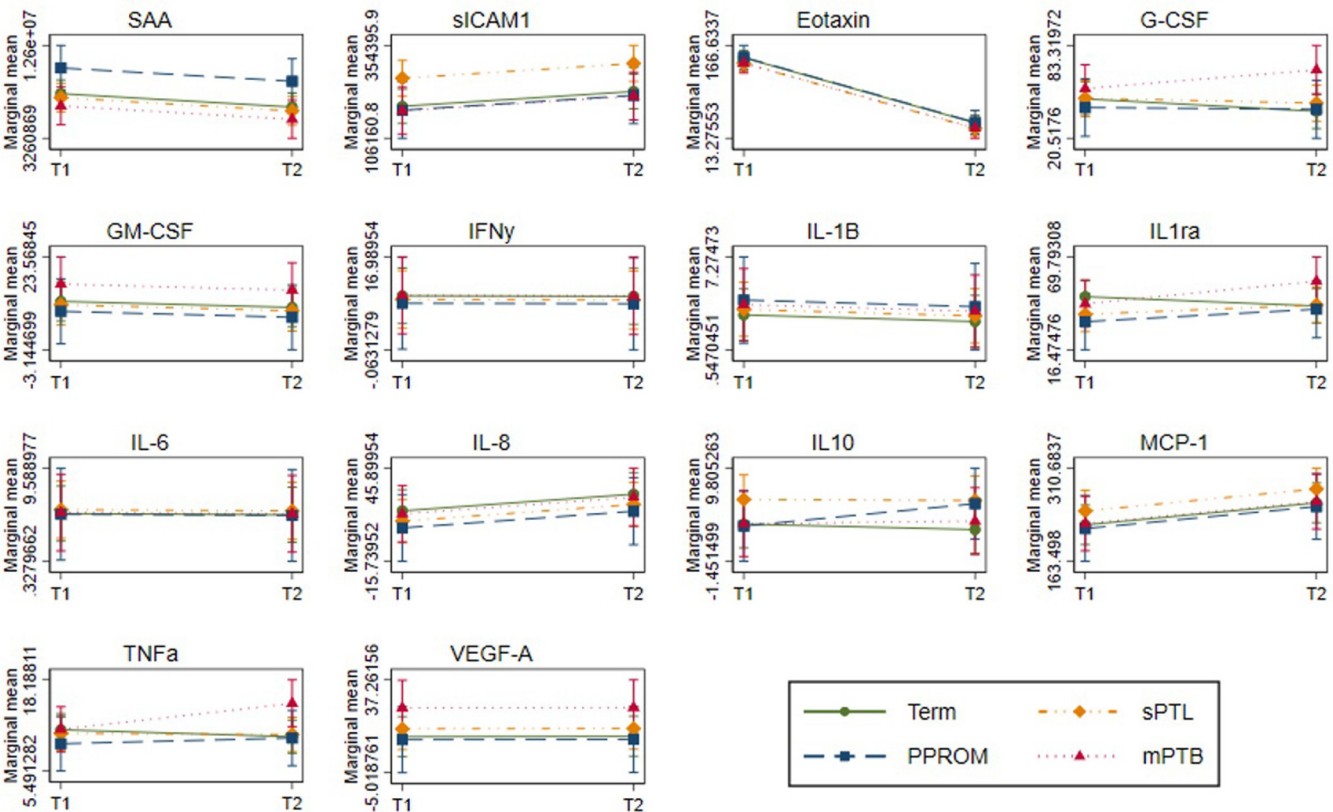

**Fig 4. The difference in mean cytokine level between time 1 and time 2 across preterm and term groups with group specific interactions.** Models were adjusted for subject specific intercepts and other maternal factors. Non-significant interaction terms between groups and time were removed. Figures were produced using the marginal model error variance structure at two timepoints in pregnancy, showing marginal mean levels of cytokine (y-axis), across two timepoints in pregnancy (x-axis).

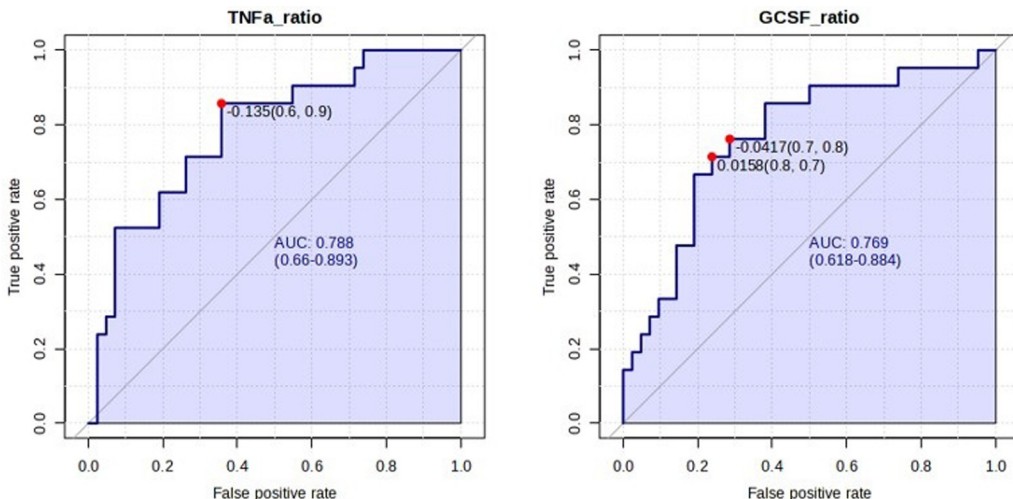

**Fig 5. ROC curve for classification with top biomarkers for mPTB.** Optimal cutoff (sensitivity, specificity) based on closest to the top left indicated with red dot. AUC and 95% confidence interval shown in blue text. AUC is significantly greater than random classification (AUC = 0.50), p<0.05.

of natural killer cells and mitochondrial dysfunction, suggesting a possible mechanism for TNFα in pregnancy-specific hypertensive disorders [44]. No biomarkers exhibited significant prediction of other preterm birth subtypes.

A previous analysis of maternal blood biomarkers using a Classification and Regression Tree (CaRT) analysis demonstrated that MCP-1, MIP-1b, eotaxin and sCD40L are associated with hypertensive disorders of pregnancies [23]. We did not observe similar associations between those biomarkers (namely, MCP-1 and eotaxin) and mPTB, despite hypertensive disorders being the most common indication for delivery in this subtype. Teasing out the similar and dissimilar cytokine and growth factor expression patterns of those pregnancies with hypertensive disorders and those with mPTB may better elucidate the interplay between these two phenomena.

## Time to delivery analysis

To determine whether cytokines could predict time to delivery, a multilevel growth curve model analysis was conducted using time to delivery (in weeks), instead of timepoint of collection. While we found a significant difference in the conditional mean level of SAA within the PPROM group when compared to the term group, the levels of SAA decrease significantly over time in all pregnancies, and those pregnancies which delivery following PPROM have a significantly higher level across all measured timepoints. However, when adjusting for a subject-specific intercept and slope, intra-subject variance accounts for 64% of the variance within the SAA model, even when conditioned on the fixed effects covariates. This limits the utility of SAA as a biomarker using this data, and further research would be required to determine baseline levels of SAA through pregnancy at multiple timepoints. As similarly found in the previous model, sICAM1a, MCP-1, and IL-8 levels increase significantly over time, and SAA, GM-CSF and Eotaxin decrease significantly over time. We found that the conditional mean level of eotaxin was significantly higher in all preterm groups as compared to the term group. Additionally, within subject variance represents almost none of the total variance (4.39e-20%), suggesting that almost all of the variance within the model is due to between-comparisons. However, eotaxin levels decrease significantly with gestational age in all groups. As the time to delivery is standardized based on delivery, rather than gestational age, this limits any clinical use of these cytokines as biomarkers for time to delivery without previous knowledge of delivery date. Group specific interactions between group and time to delivery were found for G-CSF, IL1ra, IL-10 and TNFα, similar to those interactions found between group and timepoint of collection. Summarized in S2 File.

## Summary of main findings

In this study, we characterized circulating levels of inflammatory cytokines in maternal peripheral blood at two timepoints in pregnancy during the second and third trimesters, explored their relationship with gestational age at delivery, and their potential as biomarkers for prediction of three preterm birth subtypes. We identified an association between elevated sICAM-1 levels and sPTL, though not in mPTB or PPROM, indicating, especially in the latter, that these may indeed be distinct aetiologies. Although we achieved significant classification of mPTB using cytokine levels, they are unlikely to be predictive in a clinical context. This highlights the distinct aetiologies of PTB and suggests that a more comprehensive approach is required to unravel the underlying relationship between inflammation and preterm birth subtypes.

## Strengths and limitations

A strength of our study lies in the clear definition of outcome groups, which were reviewed by three independent assessors cross-referencing questionnaire and electronic health record data.

This allowed for analysis of different preterm birth subtypes, and to demonstrate that inflammatory profiles and the relationship between inflammation and the aetiology of preterm birth likely differs by clinical presentation. The All Our Families cohort has collected extensive questionnaire and medical records as compared to comparable pregnancy cohorts, allowing for more precise classification of preterm birth subtypes. There is no consensus for what constitutes spontaneous preterm birth, as some have defined PPROM and sPTL to have shared aetiology, while others have not. Our findings suggest there may be some differences between sPTL and PPROM with respect to circulating inflammatory cytokines, as sICAM-1 was associated with the mechanism of sPTL, but not PPROM. However, clear definitions of outcome groups in future work are necessary to tease out the heterogeneity of PTB. Furthermore, while previous studies have shown associations at single time points, this was the first to explore not only multiple measurements, but the trajectory of cytokine levels through gestation. Indeed, in the case of G-CSF and TNFα, the trajectory of cytokine levels was more closely associated with pregnancy outcomes as compared to earlier or single time measurements. We also conducted a direct comparison of serum and plasma for cytokine levels, demonstrating their suitability and interchangeability for future work investigating cytokine levels in maternal blood.

This nested case control study took advantage of historical samples collected for the All Our Families pregnancy cohort study, which collected biological samples from 2008–2010. Samples were stored continuously at -80˚C for 9–11 years prior to cytokine analysis. The literature suggests that many cytokines are stable in serum and plasma at least two years [45], though very-long term stability is a gap in knowledge that remains to be addressed. Further exploration of serum and plasma cytokine biomarkers collected more recently may identify biomarkers which may not have been detected in this study due to degradation over time. However, pregnancy cohorts of this size are limited due to financial and time constraints. Nonetheless, given that both the cases and control biological samples underwent identical storage conditions, the authors suggest that patterns associated with preterm birth would remain similar.

## Heterogeneity in cytokine profiles

Associations between inflammatory cytokines in maternal blood and sPTL have been reported in pregnancy cohorts from the United States of America [9, 11, 14], Denmark [10, 19, 46], Italy [15], Korea [16], India [18], and China [47], though this is the first, to our knowledge, to be conducted in a Canadian pregnancy cohort. A systematic review identified that G-CSF was most consistently reported (in 3 of 5 studies) to be associated with sPTL [22], yet G-CSF was not associated with sPTL in our study, was only modestly associated with mPTB and not predictive. Given the previously reported associations, we sought to test predictive utility in a well-defined, relatively homogenous Canadian pregnant population. This approach also allows for exploration of the distinct molecular aetiologies of sPTL and PPROM. Though we have reported similar trends in inflammatory profiles, these are not likely to be clinically useful in this population.

## Conclusion

This data suggests that cytokine profiles do not distinguish term from preterm delivery. Whether inflammation plays a causative role in labour remains to be seen. Moving forward, it is imperative to narrow our focus on promising novel biomarkers–which, maternal peripheral blood inflammatory cytokines are likely not. The use of cell-free markers of uterine origin [48], other biological fluids which may reflect the uterine environment more closely, such as cervicovaginal fluid [49], or exploring pathways involved in the process of labour may better identify biomarkers better suited to the prediction of preterm birth. Given this, it is also

important to consider the complex phenotypes of preterm birth which likely involve distinct mechanisms, each of which warrants exploration. This likely begins with a better understanding of the mechanisms of labour, which, in turn, can advise future prediction strategies.

## Supporting information

**S1 Table. The difference in mean cytokine level between timepoint 1 and timepoint 2 in term and preterm groups, with group specific interaction terms.** Models are adjusted for subject specific intercepts and other maternal factors. Values are represented as slope between timepoints and 95% confidence intervals. Nonsignificant interactions between timepoint and group were removed from the models.
(DOCX)

**S1 File. Cytokine levels in matched serum and plasmas at the same mid-gestation timepoint.** Serum and plasma samples as part of a secondary analysis of a proof of principle study were analysed to validate the use of either biological fluid for measurement of cytokines.
(PDF)

**S2 File. Time to delivery (weeks) analysis.** Crude and adjusted regression with subject-specific intercepts and slopes with time to delivery in weeks, as well as analysis to identify group specific interactions.
(DOCX)

## Author Contributions

**Conceptualization:** Kylie K. Hornaday, Nikki L. Stephenson, Donna M. Slater.

**Data curation:** Kylie K. Hornaday, Nikki L. Stephenson, Mary T. Canning, Suzanne C. Tough, Donna M. Slater.

**Formal analysis:** Kylie K. Hornaday, Nikki L. Stephenson.

**Funding acquisition:** Suzanne C. Tough, Donna M. Slater.

**Investigation:** Kylie K. Hornaday.

**Methodology:** Kylie K. Hornaday, Nikki L. Stephenson.

**Project administration:** Mary T. Canning.

**Supervision:** Suzanne C. Tough, Donna M. Slater.

**Visualization:** Kylie K. Hornaday.

**Writing – original draft:** Kylie K. Hornaday.

**Writing – review & editing:** Kylie K. Hornaday, Nikki L. Stephenson, Suzanne C. Tough, Donna M. Slater.

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
