## [Decision Letter · Decision Letter 0]

20 May 2024

PONE-D-24-10221Maternal cytokine profiles in second and early third trimester are not predictive of preterm birthPLOS ONE

Dear Dr. Slater,

Thank you for submitting your manuscript to PLOS ONE. After careful consideration, we feel that it has merit but does not fully meet PLOS ONE’s publication criteria as it currently stands. Therefore, we invite you to submit a revised version of the manuscript that addresses the points raised during the review process.

**ACADEMIC EDITOR: **

**Both reviewers submitted favorable comment. Please revise adequately following their advice.**

We look forward to receiving your revised manuscript.

Kind regards,

Kazumichi Fujioka

Academic Editor

PLOS ONE

 [The All Our Families study was funded by an Alberta Innovates Interdisciplinary Team Grant #200700595 and Alberta Children’s Hospital Foundation. Cytokine analysis was funded by CIHR (Project funding #PJT-173295)].  

3. In the online submission form, you indicated that [Data cannot be shared publicly as the ethics agreement and participant consent from the original All Our Families Cohort study does not allow for the data to be publicly available. Data are available upon request from the University of Calgary (allourfamilies@ucalgary.ca) for researchers who meet the criteria for access to confidential data.]. 

4. lease review your reference list to ensure that it is complete and correct. If you have cited papers that have been retracted, please include the rationale for doing so in the manuscript text, or remove these references and replace them with relevant current references. Any changes to the reference list should be mentioned in the rebuttal letter that accompanies your revised manuscript. If you need to cite a retracted article, indicate the article’s retracted status in the References list and also include a citation and full reference for the retraction notice.

Additional Editor Comments:

Reviewers' comments:

Reviewer's Responses to Questions

**Comments to the Author**

1. Is the manuscript technically sound, and do the data support the conclusions?

Reviewer #1: Yes

Reviewer #2: Yes

2. Has the statistical analysis been performed appropriately and rigorously? 

Reviewer #1: Yes

Reviewer #2: Yes

3. Have the authors made all data underlying the findings in their manuscript fully available?

Reviewer #1: Yes

Reviewer #2: Yes

4. Is the manuscript presented in an intelligible fashion and written in standard English?

Reviewer #1: Yes

Reviewer #2: Yes

5. Review Comments to the Author

Reviewer #1: This study is a well designed and scientifically sounds good.

the methodology is ok.

the studied cytokines were well described. the statistical analysis cover the discussion and main conclusion of the study.

However, i would suggest one minor suggestion that:

1) is this a retrospective or prospectively organized study? the blood samples were taken and stored at -80 degree for future investigation. this should be clarified in the first paragraph of material and method section.

Reviewer #2: Thank you for the possibility to review the very interesting manuscript. The idea of the study is very important and actual, especially the investigations in different groups of preterm birth.

I suggest to analyze the duration from the sampling till the delivery and then to check the prediction of preterm delivery in one, two or three weeks, may be the results could be different. Also I suggest to describe the stability of cytokines and freezing / thawing (the samples were stored at -80 0C prior to cytokine expression analysis).

Other recommendations:

- the incidence of preterm delivery is cited from the publication in 2019, I suggest to cite the newer one (lines 47-49).

- the calculations should be reevaluated in Tables (Table 1 - education - sPTL - 13+72+13, it would be 98, but not 100 percent. Other - etnicity, Table 2 - mode of delivery, labour, reason for caesarean delivery etc.)

6. PLOS authors have the option to publish the peer review history of their article (what does this mean?). If published, this will include your full peer review and any attached files.

Reviewer #1: No

Reviewer #2: **Yes: **Diana Ramasauskaite

---

## [Author Response · Author response to Decision Letter 0]

26 Jul 2024

Dear academic editor and reviewers,

Thank you for your thoughtful comments and feedback on our submitted manuscript. The authors have made the following changes to the manuscript in response to your review.

1. The manuscript has been checked to ensure it meets PLOS ONE’s style requirements, including renaming the file to “Manuscript” as required.

2. The financial disclosure statement to state that the funders had no role in study design, data collection and analysis, decision to publish, or preparation of the manuscript.

3. The data cannot be made publicly available as agreed by the original ethics approved by the Conjoint Health Region Ethics Board (CHREB) at the University of Calgary and consent, as public availability would compromise patient privacy. However, data is made available on request from the University of Calgary (allourfamilies@ucalgary.ca) following proposal submission and ethics review for researchers who meet the specific criteria for access to confidential data

4. The reference list has been reviewed and no references have been retracted to the best of the authors’ knowledge.

Reviewer comments:

Reviewer #1: This study is a well designed and scientifically sounds good.

the methodology is ok.

the studied cytokines were well described. the statistical analysis cover the discussion and main conclusion of the study.

However, i would suggest one minor suggestion that:

1) is this a retrospective or prospectively organized study? the blood samples were taken and stored at -80 degree for future investigation. this should be clarified in the first paragraph of material and method section.

Response to Reviewer #1:

Thank you to this reviewer for the kind feedback. We have clarified that the All Our Families cohort study was a prospective study and included this information in line 87 of the manuscript, in the first paragraph of the materials and methods section as suggested.

Reviewer #2: Thank you for the possibility to review the very interesting manuscript. The idea of the study is very important and actual, especially the investigations in different groups of preterm birth.

I suggest to analyze the duration from the sampling till the delivery and then to check the prediction of preterm delivery in one, two or three weeks, may be the results could be different. Also I suggest to describe the stability of cytokines and freezing / thawing (the samples were stored at -80 0C prior to cytokine expression analysis).

Other recommendations:

- the incidence of preterm delivery is cited from the publication in 2019, I suggest to cite the newer one (lines 47-49).

- the calculations should be reevaluated in Tables (Table 1 - education - sPTL - 13+72+13, it would be 98, but not 100 percent. Other - etnicity, Table 2 - mode of delivery, labour, reason for caesarean delivery etc.)

Response to Reviewer #2:

Thank you to this reviewer for the kind feedback and comments. Time to delivery analysis was added as suggested by the reviewer (lines 342-363, S2 File). We first attempted to incorporate time to delivery (in weeks) as part of the original model. Unfortunately, due to the study design, in which biosamples were collected at two specific timepoints in pregnancy, the association between the group and the time to delivery was strong (e.g. RR for sPTL = -34; p<0.0001). Correlation between exposure, collection time, and time to delivery is high, which violates the regression assumption that the exposure and covariates are statistically independent. Instead, the collection time was removed from the regression and replaced with time to delivery to satisfy the assumption of independence. While we did see some associations between mean cytokine levels and time to delivery, this analysis required prior knowledge of delivery date, limiting the utility of these biomarkers to predict time to delivery. Future work with appropriate study design and sample collection time is suggested to better identify biomarkers for predicting time to delivery. In the interest of further exploring possible associations within groups, we additionally included group specific interactions between group and a) timepoint of collection (lines 281-306, Table S1) and b) time to delivery in weeks (S2 file), which indicates that the rate of change of mean cytokine levels of G-CSF, IL1ra, IL10 and TNFa vary between groups. To summarize, G-CSF, IL1ra and TNFa levels increase with time in mPTB, but remain the same or decrease in other groups. On the other hand, IL-10 levels increase in those PTB following PPROM, whereas levels remain the same in other groups. We thank this reviewer for their thoughtful comments and hope this additional analysis will provide more insight into the possible associations with preterm birth.

We appreciate that the long-term storage of these blood samples may be a possible limitation of this study. Samples were stored at -80oC for 9-11 years and did not undergo freeze thaw cycles prior to cytokine analysis. All cytokine analyses was performed at the same time so that group comparisons could be made. Unfortunately, we do not have data available for serum and plasma cytokine content closer to sample collection, though literature as shown strong stability of cytokines in serum and plasma for at least 2 years. Further details on storage were added to lines 98-99, and a discussion of this limitation was added to lines 395-404. 

The incidence of preterm delivery was updated to the most recent data available, lines 48-49:

Ohuma EO, Moller AB, Bradley E, Chakwera S, Hussain-Alkhateeb L, Lewin A, et al. National, regional, and global estimates of preterm birth in 2020, with trends from 2010: a systematic analysis. Lancet. 2023;402(10409):1261-71.

The values in Tables 2 and 3 have been reviewed and adjusted as appropriate, including indicating values for missing data. Totals should now all equal 100% (+/- 1 to account for rounding) for appropriate categories. 

Thank you again to the reviewers and editor for their support and feedback in this work. We look forward to hearing your response to our revisions.

Kind regards,

Donna M Slater, 

On behalf of the authors: Kylie K Hornaday, Nikki L Stephenson, Mary T Canning, Suzanne C Tough

---

## [Editor Report · Decision Letter 1]

24 Sep 2024

Maternal cytokine profiles in second and early third trimester are not predictive of preterm birth

PONE-D-24-10221R1

Dear Dr. Slater,

We’re pleased to inform you that your manuscript has been judged scientifically suitable for publication and will be formally accepted for publication once it meets all outstanding technical requirements.

Kind regards,

Kazumichi Fujioka

Academic Editor

PLOS ONE

Additional Editor Comments (optional):

Well revised.
---

## [Editor Report · Acceptance letter]

1 Oct 2024

PONE-D-24-10221R1 

PLOS ONE

Dear Dr. Slater, 

I'm pleased to inform you that your manuscript has been deemed suitable for publication in PLOS ONE. Congratulations! Your manuscript is now being handed over to our production team.

Kind regards, 

on behalf of

Dr. Kazumichi Fujioka 

Academic Editor

PLOS ONE